## Correspondence

# Multimodal AI agents for capturing and sharing proteomics laboratory practice

Patricia Skowronek [1], Anant Nawalgaria[2] & Matthias Mann [1✉]

Much of a scientist's expertise is learned through hands-on practice, not from manuals. This implicit knowledge—the subtle variations to a protocol or the instinct for troubleshooting—is critical in technique-intensive domains like our field of mass spectrometry (MS)-based proteomics, yet it is rarely documented. As chemist and philosopher Michael Polanyi observed, "we can know more than we can tell" (Polanyi, 1966). This challenge is amplified by the constant turnover in academic labs, weakening reproducibility and making cutting-edge science less accessible. Existing automation helps, but most systems are rigid and limited to liquid handling, offering little flexibility for high-pace research.

Generative AI promises to lower barriers to scientific discovery. Large language models can already review literature, analyze bioinformatics data, and even orchestrate multi-step research workflows—from generating hypotheses and writing code to analyzing results and drafting manuscripts (Yao et al, 2023; Lobentanzer et al, 2025; Liu et al, 2025; Miao et al, 2025; Schmidgall et al, 2025; Gottweis et al, 2025; Roohani et al, 2025; Swanson et al, 2025). In life science, this concept is beginning to extend into the physical lab, where AI has been used to program liquid-handling robots (Boiko et al, 2023).

Yet these tools still do not address the written vs. tacit knowledge divide—they do not support researchers in their hands-on laboratory tasks. Here, we demonstrate an AI agent that uses video to capture, share, and apply expert knowledge. By linking digital protocols with physical laboratory work, this system provides scalable, personalized guidance on complex procedures and offers a path toward more accessible, reproducible science.

To provide this guidance, we designed the AI agent to mimic an experienced colleague—able to answer questions, spot errors, and draw on years of practical know-how. The agents' responses, adapted to current equipment conditions and researcher skill level, are delivered through a multi-agent AI framework that is accessed via a chat interface.

Our system is built using Google's Agent Development Kit (ADK), Gemini 2.5 models accessed through Google Cloud, and the Model Context Protocol (MCP) for tool integration. We selected Gemini 2.5 for its multimodal capabilities and long context window, which enables processing of multiple videos within one prompt, and the Agent Development Kit (ADK) for its specialized multi-agent orchestration with Gemini models. A main agent interprets a researcher's request and delegates it to specialized sub-agents (Fig. 1A). Each sub-agent has a specific function. For example, a Protocol Agent analyzes a video of a person performing a specific sequence of actions to generate a protocol. Given an existing one, a Lab Note Agent detects errors or omissions when a researcher carries out the procedure. Other agents connect to the laboratory's resources: the Lab Knowledge Agent retrieves documents from an internal knowledge base, the Instrument Agent monitors the performance of mass spectrometers, and a QC Memory Agent logs all quality control ratings to a central database, preserving the troubleshooting history for the entire team. These agents are augmented with custom prompts encompassing a diverse set of examples and a proteomics knowledge base, enabling them to optimally reason through tasks. We evaluated the system's performance after creating a benchmark dataset of lab protocols and videos (see Appendix).

To test the system in practice, we consider a common question when operating sophisticated analytical instruments: "Is this instrument ready for use?" To answer this, the agent goes beyond the latest quality control (QC) metrics, benchmarking current performance against historical data and expert decisions to issue a recommendation. It then records the researcher's subsequent actions and supplies the necessary protocols for the next step (Appendix Fig. S1).

This process contributed a continuously improving knowledge base, thereby building institutional memory that captures a specific form of otherwise lost expert reasoning, in this case for mass spectrometer operation based on QC metrics.

A key aspect of capturing expert knowledge is documenting procedures. Yet, writing informative, detailed, and reproducible experimental protocols is a time-consuming process, which is often neglected. An AI can help in generating protocols using its expert knowledge, but in our experience, this may involve invented details or hallucinations. Our agent avoids this by a video-to-protocol approach, grounding the process in direct observation. In practice, a researcher simply records their actions while explaining the procedures. Subsequently, the protocol agent analyzes the visual and audio data using its supplemented proteomics expertise, capturing the interactions with equipment, and generating a formatted protocol in minutes (Appendix Fig. S2).

We compared the AI-generated protocols across ten diverse procedures, from simple pipetting to complex mass spectrometer operations, to an established lab ground-truth. Human raters assigned the AI protocols an average quality score of 4.1 out of 5 based on their completeness, technical accuracy, logical flow, safety instructions, and formatting (see Appendix and Appendix Fig. S3). Overall, we found

[1]Department of Proteomics and Signal Transduction, Max Planck Institute of Biochemistry, Martinsried, Germany. [2]Google, Munich, Germany. ✉E-mail: mmann@biochem.mpg.de

https://doi.org/10.1038/s44320-025-00179-1 | Published online: 15 December 2025

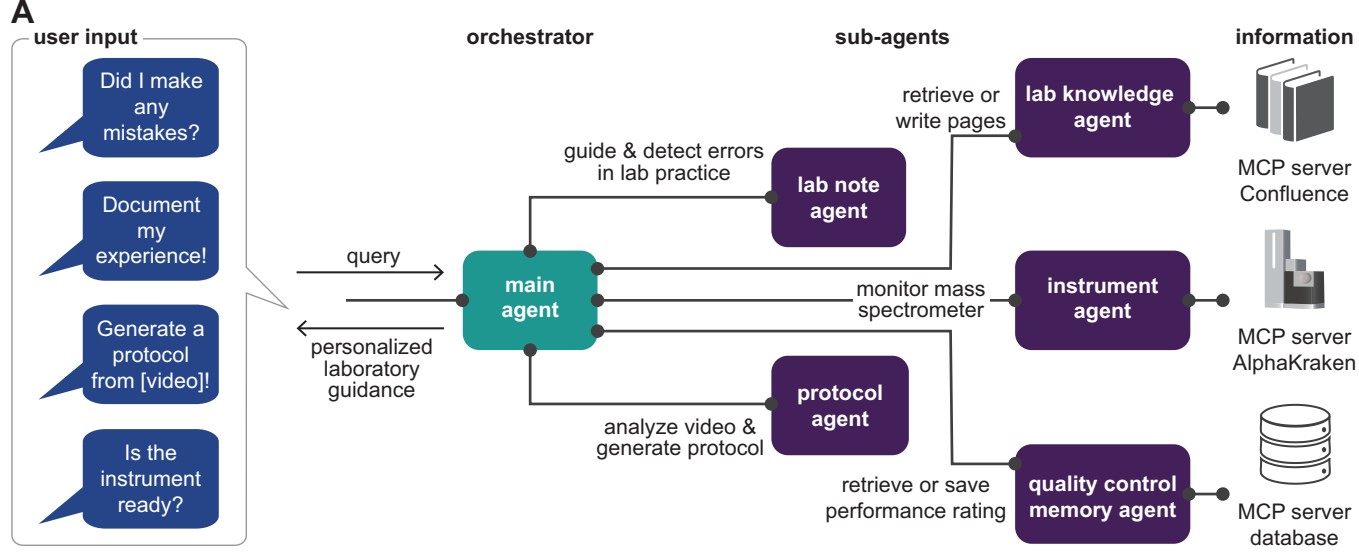

**Example: Error detection during hands-on laboratory experiment:**

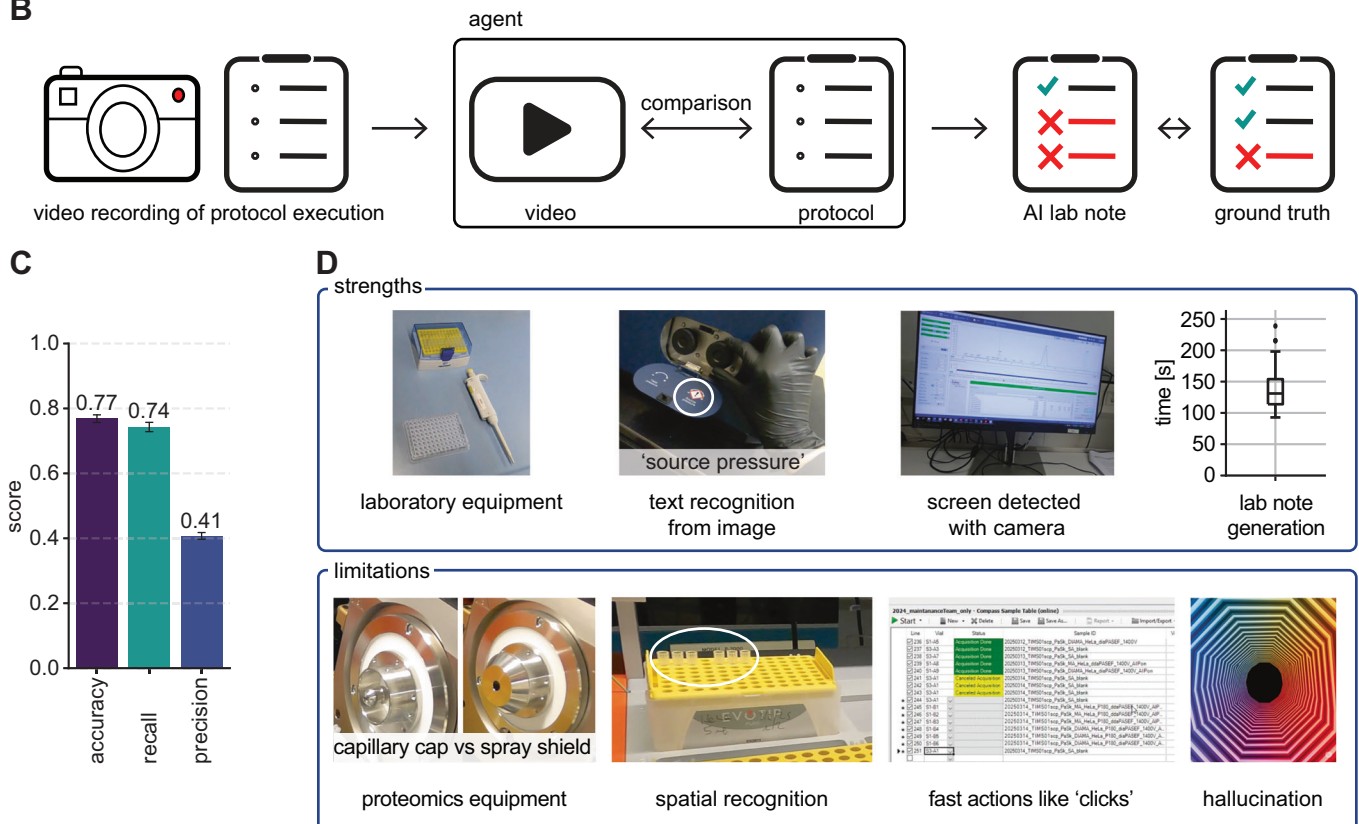

that AI-assisted protocol generation saves time while also creating detail-rich instructions that surpass the brief method sections of scientific papers.

Once the AI is equipped with a protocol, it can in turn observe a researcher's lab actions and point out omissions or

mistakes. This should be especially helpful for researchers in training. To test this, we created a benchmark dataset by recording a scientist intentionally making 70 mistakes across 28 experiments (421 total steps).

We then investigated whether our Lab Note agent could detect errors and

omissions by comparing a researcher's actions on video to a reference protocol and by flagging deviations (Fig. 1B; Appendix Fig. S4). Compared to ground-truth, the agent correctly identified three-fourths of all mistakes, achieving an accuracy of 0.77 and a recall of 0.74 (Fig. 1C). While

◀ **Figure 1. Proteomics lab agent for multimodal guidance.**

(A) The agentic framework translates conversational queries into personalized laboratory guidance. A main agent orchestrates specialized sub-agents: the Protocol Agent generates protocols from videos; the Lab Note Agent detects experimental errors; the Lab Knowledge Agent manages a knowledge base; the Instrument Agent monitors mass spectrometers; and the Quality Control Memory Agent accesses historical performance data. (B) An experimental workflow realized in our laboratory. A workflow where a scientist performed a protocol with intentional errors while recording the process. The AI agent analyzes the video against the baseline protocol to generate a lab note with flagged deviations. The agent's performance was evaluated by comparing its AI lab note to a manually annotated lab note (ground-truth). (C) Overall agent performance metrics on the benchmark dataset, averaged over triplicate runs (error bars indicate standard deviation). (D) Examples of the agent's strengths and limitations. Strengths include recognizing standard laboratory equipment, text on equipment, on-screen computer actions, and rapidly generating laboratory notes. Limitations include difficulty distinguishing visually similar, specialized proteomics equipment, achieving fine-grained spatial recognition (e.g., specific wells), detecting actions faster than 1 Hz video sampling rate (e.g., mouse clicks), and occasionally hallucinating events. Source data are available online for this figure.

precision was lower (0.41), indicating some false positives, a higher recall is preferable in this setting; it is better to flag a potential error for review than to miss a critical one.

Analysis of the AI error detection process showed that the agent effectively identified general equipment and on-screen text, but struggled with tasks requiring fine-grained spatial recognition, domain-specific knowledge, and the detection of fast actions (Fig. 1D, Appendix Fig. S5). The latter accounted for most false positive identifications and explains why the agent successfully identified most omitted steps (91% recall) but missed subtle errors in execution (53% recall) or changes in step order (38% recall) (Appendix Fig. S5D). Generating lab notes on average took just over two minutes and less than a dollar in token costs. These promising initial results can clearly be improved further as the abilities of multimodal agents advance fast, a trend we already experienced during development when upgrading to the more capable model Gemini 2.5 Pro, and can be further accelerated through domain-specific fine-tuning.

This study shows that our multimodal agent-based approach can transfer practical expertise by linking written instructions with proteomics laboratory work through multimodal understanding of video, speech, and text. The approach should be adaptable to scientific fields other than proteomics that also rely on complex, hands-on procedures.

In the near term, such systems can provide real-time, interactive guidance that prevents errors before they happen and makes experiments more efficient. Over time, they can build an institutional memory that preserves expertise beyond individual researchers, standardize protocols, and even support predictive maintenance by tracking both instrument parameters and human actions.

Once this system is fully implemented, scientists will literally be able to talk directly to these agents, receiving interactive guidance without having to stop and consult a written protocol.

This approach helps automate the routine parts of proteomics workflows that benefit from standardization, while preserving the human expertise needed for complex decisions. However, it is not a vision of blind automation in which tacit and artisanal knowledge is lost, but rather a way to capture and share it.

So far, we have validated the agent's performance only in our own laboratory, meaning that the system's prompts and architecture were developed within the same context used for performance evaluation. As a next step, it will be interesting to evaluate its generalizability across diverse laboratory environments, researcher practices, and models.

Regarding "institutional memory", a key contribution of our framework is providing the mechanism to capture multifaceted knowledge in a structured format. In this context, some components—like expert reasoning on specific instrument QC—are naturally laboratory-specific, as instrumentation and procedures often differ between groups. Other components, however, such as the video-generated protocols and lab notes, are inherently more shareable, and the collected data may be useful in developing predictive maintenance models that can be fine-tuned for other settings.

Realizing this potential to democratize science by spreading best practices requires a shared commitment. To this end, this study contributes a public benchmark dataset for evaluating AI on complex tasks in the high-barrier scientific domain of proteomics. Its further development should be a community effort, embedding shared standards for transparency and supervision so that these tools augment rather than replace the rigor of scientific practice. If

achieved, such AI partners could fundamentally transform laboratory work.

## Data availability

Complete code for agent and analysis is available on GitHub and the benchmark dataset consisting of videos, protocols and ground-truth lab notes is available on Zenodo at https://doi.org/10.5281/zenodo.17253028.

## Peer review information

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

## Acknowledgements

This study was supported by the Max Planck Society for the Advancement of Science. We thank our colleagues at the Department of Proteomics and Signal Transduction for their support and fruitful discussions. We are grateful for the input and constant support throughout the entire project from Magnus Schwörer and Fabian Opitz.

## Author contributions

**Patricia Skowronek**: Conceptualization; Resources; Data curation; Formal analysis; Validation; Investigation; Visualization; Methodology; Writing —original draft; Writing—review and editing. **Anant Nawalgaria**: Conceptualization; Resources; Software; Validation; Methodology. **Matthias Mann**: Conceptualization; Supervision; Funding acquisition; Writing—original draft; Project administration; Writing—review and editing.

## Disclosure and competing interests statement

AN is an employee of Google. MM is an indirect investor in Evosep Biosystems and OmicsVision. MM is an editorial advisory board member. This has no bearing on the editorial consideration of this article for publication. The remaining authors declare no competing interests.

