## [Peer Review File · Molecular Systems Biology]

Multimodal AI agents for capturing and sharing proteomics laboratory practice

Patricia Skowronek, Anant Nawalgaria, and Matthias Mann

Corresponding author(s): Matthias Mann (mmann@biochem.mpg.de)

Review Timeline:

Submission Date:	3rd Oct 25
Editorial Decision:	22nd Oct 25
Revision Received:	29th Nov 25
Accepted:	2nd Dec 25

Editor: Jingyi Hou

Transaction Report:

22nd Oct 2025

Manuscript Number: MSB-2025-13390
Title: Multimodal AI agents for capturing and sharing laboratory practice
Author: Patricia Skowronek
Anant Nawalgaria
Matthias Mann

Dear Matthias,

Thank you for submitting your manuscript. We have now received the reviewer's report. As you can see below, the reviewer finds the work timely and well-executed and recommends acceptance pending relatively minor revisions. However, a few key issues must be addressed to further strengthen the manuscript.

Without reiterating all the points raised in the review, some of the more substantial issues are the following:

1. The reviewer raises important concerns regarding the generalizability of the proposed framework. The claims about general laboratory practice should be narrowed to reflect the specific domain (e.g., proteomics) in the title and conclusion. Additionally, the reviewer is concerned about whether the system's knowledge representation is transferable and robust to deployment across different contexts, which needs to be carefully discussed.

2. The technical questions related to the LLM need to be addressed.

On a more editorial level:

1. Please include up to five keywords in the manuscript file.

2. The figure should be removed from the manuscript file and uploaded as a separate figure file. The figure legend should be placed at the end of the manuscript, after the References section.

3. Funding information should be included in the "Acknowledgements" section and must be consistent with the information provided in the online submission system.

4. Please ensure the sections are ordered as follows: Title Page - Acknowledgements - Disclosure and Competing Interests Statement - References - Figure Legend.

5. The "Supplementary Material" section should be removed from the manuscript file.

6. Appendix: The title page should read "Appendix for [Manuscript Title]" instead of "Supplementary Material for...".

Additional information on source data and instruction on how to label the files are available

8. The section titled "Potential conflicts of interest" should be renamed to "DISCLOSURE AND COMPETING INTERESTS STATEMENT." Please also add the following sentence:

"MM is an editorial advisory board member. This has no bearing on the editorial consideration of this article for publication."

Please attach a covering letter giving details of the way in which you have handled each of the points raised by the referees. A revised manuscript may be once again subject to review and you probably understand that we can give you no guarantee at this stage that the eventual outcome will be favorable.

I look forward to receiving a revised manuscript soon.

Kind regards,
Jingyi

Jingyi Hou, PhD

*** PLEASE NOTE *** As part of the EMBO Press transparent editorial process initiative (see our Editorial at <https://dx.doi.org/10.1038/msb.2010.72>), Molecular Systems Biology publishes online a Review Process File with each accepted manuscripts. This file will be published in conjunction with your paper and will include the anonymous referee reports, your point-by-point response and all pertinent correspondence relating to the manuscript. If you do NOT want this File to be published, please inform the editorial office at contact@molsystbiol.org within 14 days upon receipt of the present letter.

Reviewer #1:

Skowronek et al. present a framework and benchmark for the assistance of proteomics laboratory practice using agentic AI. The approach is timely and implemented in a solid fashion. I can recommend acceptance after a limited revision regarding the points below.

My main question is whether this approach, which is limited along several axes (domain, location of experiments, model and software stack), is generalisable and thus warrants some of the relatively general claims.

The application domain (laboratory environment) is very narrow (mass spectrometry proteomics); while the authors are an authority in this particular field, the manuscript in my opinion cannot make claims that are related to more general laboratory practice, as the system's performance may differ across the many domains of laboratory science. I would propose to specify "proteomics laboratory" or "mass spectrometry laboratory" in text that refers to general laboratory practice, such as the title and conclusion.

Relatedly, it seems that the baseline and benchmark recordings stem from the same "in-distribution" location in Munich; as such, it is not clear how much of the final performance can be attributed to the fact that the development and testing of the system happened in the same conditions. While model training or fine-tuning does not seem to have happened, the prompts and architecture used in the system were developed in the very same context that was also used to measure performance. Some risk of leakage has to be assumed. It would be more convincing if the benchmark included a test set of recordings with similar tasks performed in a different lab by different researchers. This seems particularly relevant in light of the authors' comments that the future work on the framework should be a community effort. At least, this limitation should be addressed in the text.

No justification is given for the choice to use Gemini and the ADK, although it is not difficult to guess the reason. Nevertheless, this bias / limitation should be discussed in the context of the conclusions drawn, and general conclusions of what "AI" can do (e.g., "AI can transfer practical expertise") should be appropriately contextualised to what has actually been shown. I do not think a comprehensive comparison to other model vendors is required for the advance that this manuscript describes, but not every user in the community may want or be able to use the particular models and ADK used by the authors.

In extension, it has not been shown that this system can capture "otherwise lost expert reasoning" in general. It is not clear how transferable this expert reasoning is between contexts, or if it "only" serves as a helpful device in saving time in the lab where it was established. It is not obvious how generalised this "institutional knowledge" can or should be; for instance, in the given case, is it reasonable or achievable to deploy the solution as a single instance for the MPI Martinsried, or across the department for proteomics? How will the expert knowledge be captured to ensure compatibility between different groups and with future recordings? The SQL schema seems ad hoc and does not pass a FAIR assessment (e.g., with respect to unambiguous identifiers for classes and records). What happens if deployments in other labs come up with subtly different uses of the schema / interpretations of the database agent prompt? Will discrepancies be detected?

Details

I do not understand the reasoning in this sentence in the Appendix-"We avoided a temperature of 0, which often led to repetitive outputs." What is problematic about repetitive outputs in this case? What is the difference between repetitive and reproducible? If the models' task is only to decide in a binary fashion between correct and incorrect, why does "repetitiveness" matter? Is this a problem with the zero-shot behaviour of the Gemini model, specifically; i.e., are the top1 token probabilities not always suitable for this this task? Should the model be fine-tuned for lab protocol error detection to make this step more robust?

The LLM-as-a-judge approach: was it confirmed that the model's Likert scale corresponds to an accurate semi-quantitative assessment? In other words, is the LLM judge calibrated with respect to the rubrics that were assessed (completeness, technical accuracy, etc)? A sample of LLM judgments could be re-evaluated blindly and the agreement to a human rater calculated by Cohen's Kappa or correlation.

Is the LLM-as-a-judge evaluation sufficiently general to evaluate application of the system in other labs? To what extent? Only for MS proteomics, related disciplines, less related wet lab disciplines?

Could it be an issue that the errors used in the tests were deliberately introduced? How high is the risk that errors which happen in reality will be out of distribution of the artificial errors tested here?

The code repository looks solid and passes most of the thresholds for open source projects. However, I notice a lack of testing and continuous integration, which could be problematic if the community involvement the authors call for actually does happen. Without a solid test battery, contributions are very difficult to manage. I recommend adding tests to cover the existing functionality to ensure later additions do not break things. I also recommend thinking about versioning the framework (semantic versioning if possible) for the same reasons. A dedicated documentation that is automatically built from the repository (e.g., using mkdocs material) would be a further step towards sustainability in a community-driven way. Documentation solely in the GitHub README has its limitations.

Minor

I recommend another round of grammar checks (often regarding commas); for instance, the sentence "We compared ... [,] to an established lab ground truth." is ambiguous due to a missing comma; in other places, there are unnecessary ones.

November 29, 2025 Point-by-point response to reviewer comments for “Multimodal AI agents for capturing and sharing proteomics laboratory practice”

We sincerely thank the reviewer for their time and constructive feedback, which has significantly improved our manuscript. We have addressed each point as detailed below. The issues raised mostly required textual changes and improvements in the code base.

In response to the reviewer’s comments, we have made some key changes and additions:

- We have revised the manuscript to more accurately reflect the study's scope in mass spectrometry-based proteomics.
- We performed a new calibration analysis for the 'LLM-as-a-judge' approach: Expert human raters blindly evaluated the AI-generated protocols and we compared their scores to the LLM-judge scores using Spearman's correlation. This analysis showed that the updated LLM judge is well-calibrated (new Appendix Figure S3 C and D).
- We added a paragraph to directly address the “in-distribution” limitation of the benchmark, acknowledging that the system was developed and evaluated in the same laboratory context.
- We revised the code base including the SQL schema to be more FAIR-aware and added a test suite with 93% coverage.

We believe that these revisions comprehensively address the reviewer's concerns.

Reviewer #1:

Skowronek et al. present a framework and benchmark for the assistance of proteomics laboratory practice using agentic AI. The approach is timely and implemented in a solid fashion. I can recommend acceptance after a limited revision regarding the points below.

My main question is whether this approach, which is limited along several axes (domain, location of experiments, model and software stack), is generalisable and thus warrants some of the relatively general claims.

We thank the reviewer for investing substantial time in the evaluation of our manuscript and helping us to improve it with their feedback. We are pleased that they judge our work to be ‘timely and implemented in a solid fashion’.

1. The application domain (laboratory environment) is very narrow (mass spectrometry proteomics); while the authors are an authority in this particular field, the manuscript in my opinion cannot make claims that are related to more general laboratory practice, as the system's performance may differ across the many domains of laboratory science. I would propose to specify "proteomics laboratory" or "mass spectrometry laboratory" in text that refers to general laboratory practice, such as the title and conclusion.

We do believe it will generalize as our laboratory actually routinely covers the entire molecular and cellular biology workflow as well. However, we agree that we have not actually shown this and have revised the title by adding 'proteomics laboratory' in the title and we also changed the conclusion accordingly.

Title: “Multimodal AI agents for capturing and sharing proteomics laboratory practice”

Page 2:

“This study shows that our multimodal agent-based approach can transfer practical expertise by linking written instructions with ~~real~~proteomics laboratory work through multimodal understanding of video, speech, and text. The approach should be adaptable to scientific fields other than proteomics that also rely on complex, hands-on procedures.

[...]

This approach helps automate the routine parts of ~~laboratory~~proteomics workflows that benefit from standardization, while preserving the human expertise needed for complex decisions. However, it is not a vision of blind automation in which tacit and artisanal knowledge is lost, but rather a way to capture and share it.

[...]

Realizing this potential to democratize science by spreading best practices requires a shared commitment. To this end, this study contributes a public benchmark dataset for evaluating AI on complex tasks in the ~~a~~ high-barrier scientific domain of proteomics. Its further development should be a community effort, embedding shared standards for transparency and supervision so that these tools augment rather than replace the rigor of scientific practice. If achieved, such AI partners could fundamentally transform laboratory work.”

2. Relatedly, it seems that the baseline and benchmark recordings stem from the same "in-distribution" location in Munich; as such, it is not clear how much of the final performance can be attributed to the fact that the development and testing of the system happened in the same conditions. While model training or fine-tuning does not seem to have happened, the prompts and architecture used in the system were developed in the very same context that was also used to measure performance. Some risk of leakage has to be assumed. It would be more convincing if the benchmark included a test set of recordings with similar tasks performed in a different lab by different researchers. This seems particularly relevant in light of the authors' comments that the future work on the framework should be a community effort. At least, this limitation should be addressed in the text.

We agree that this is a limitation at this moment. We have added a paragraph to directly address the "in-distribution" limitation.

Page 3:

"So far, we have validated the agent's performance only in our own laboratory meaning that the system's prompts and architecture were developed within the same context used for performance evaluation. As a next step, it will be interesting to evaluate its generalizability across diverse laboratory environments, researcher practices, and models."

3. No justification is given for the choice to use Gemini and the ADK, although it is not difficult to guess the reason. Nevertheless, this bias / limitation should be discussed in the context of the conclusions drawn, and general conclusions of what "AI" can do (e.g., "AI can transfer practical expertise") should be appropriately contextualised to what has actually been shown. I do not think a comprehensive comparison to other model vendors is required for the advance that this manuscript describes, but not every user in the community may want or be able to use the particular models and ADK used by the authors.

We added a justification for our choice of Gemini and ADK so that future readers can evaluate for themselves whether another model might be better suited for their needs. Accordingly, we also contextualized what our system, rather than AI in general, can accomplish.

Page 1:

"Our system is built using Google's Agent Development Kit (ADK), Gemini 2.5 models accessed through Google Cloud and the Model Context Protocol (MCP) for tool integration. We selected Gemini 2.5 for its multimodal capabilities and long context window, which enables processing of multiple videos within one prompt, and the Agent Development Kit (ADK) for its specialized multi-agent orchestration with Gemini models. [...]"

Page 2:

"This study shows that ~~At~~ our multimodal agent-based approach can transfer practical expertise by linking written instructions with proteomics laboratory work through multimodal understanding of video, speech, and text. The approach is adaptable to other scientific fields than proteomics that also rely on complex, hands-on procedures."

4. In extension, it has not been shown that this system can capture "otherwise lost expert reasoning" in general. It is not clear how transferable this expert reasoning is between contexts, or if it "only" serves as a helpful device in saving time in the lab where it was established. It is not obvious how generalised this "institutional knowledge" can or should be; for instance, in the given case, is it reasonable or achievable to deploy the solution as a single instance for the MPI Martinsried, or across the department for proteomics? How will the expert knowledge be captured to ensure compatibility between different groups and with future recordings?

We thank the reviewer for raising this important point about the scope and generalizability of the captured knowledge. We now clarify that our 'expert reasoning' example is specific to mass spectrometry and better define the concept of 'institutional memory'.

Page 2:

"To test the system in practice, we consider a common question when operating sophisticated analytical instruments: "Is this instrument ready for use?" To answer this, the agent goes beyond the latest quality control (QC) metrics, benchmarking current performance against historical data and expert decisions to issue a recommendation. It then records the researcher's subsequent actions and supplies the necessary protocols for the next step (Appendix Figure S1).

This process contributed a continuously improving knowledge base, thereby building institutional memory that captures a specific form of otherwise lost expert reasoning, in this case for mass spectrometer operation based on QC metrics."

Page 3:

"Regarding 'institutional memory,' a key contribution of our framework is providing the mechanism to capture multifaceted knowledge in a structured format. In this context, some components - like expert reasoning on specific instrument QC - are naturally laboratory - specific, as instrumentation and procedures often differ between groups. Other components, however, such as the video-generated protocols and lab notes, are inherently more shareable and the collected data may be useful in developing predictive maintenance models that can be fine-tuned for other settings."

The SQL schema seems ad hoc and does not pass a FAIR assessment (e.g., with respect to unambiguous identifiers for classes and records). What happens if deployments in other labs come up with subtly different uses of the schema / interpretations of the database agent prompt? Will discrepancies be detected?

We thank the reviewer for this important observation. We have now revised the schema to be more robust by incorporating UUIDs for unique records, schema versioning for provenance, and renaming key fields to align with canonical standards. Interoperability and potential discrepancies are managed directly by our agentic architecture, as the QC Memory Agent now acts as the single, standardized gateway to the database. The semantic documentation for this agent is embedded in docstrings, which define the rules for agent-database interaction. Our system maintains consistency within a lab by programmatically passing identifiers like instrument_id from the instrument agent via AlphaKraken. AlphaKraken is our in-house developed tool used for automated data processing and monitoring of mass spectrometry experiments, and is freely available on GitHub. We note that other QC systems can, in

principle, be integrated if they provide an MCP server with the same API. For full reusability, our database's expert decision entries must be shared in conjunction with their corresponding AlphaKraken performance metrics. This agent-enforced framework, now with improved field standards and a clear data dependency, provides a FAIR-aware foundation. To ensure transparency, we have incorporated this rationale and the schema documentation directly into the GitHub repository accompanying this paper at create_db.py and in the documentation.

Details

5. I do not understand the reasoning in this sentence in the Appendix-"We avoided a temperature of 0, which often led to repetitive outputs." What is problematic about repetitive outputs in this case? What is the difference between repetitive and reproducible? If the models' task is only to decide in a binary fashion between correct and incorrect, why does "repetitiveness" matter? Is this a problem with the zero-shot behaviour of the Gemini model, specifically; i.e., are the top1 token probabilities not always suitable for this task? Should the model be fine-tuned for lab protocol error detection to make this step more robust?

We thank the reviewer for this helpful comment, which identified an ambiguity. The reviewer is correct that for a single-token task ('correct' or 'error'), a temperature of 0.0 would be ideal. In our analysis, however, the model generates an entire lab note, which is a multi-token task where a temperature of 0.0 could cause undesirable generation loops. We have revised the manuscript paragraph to make this distinction clear (see below) and have also added a complete example in the Appendix (Page 13-16) to make the process of lab note generation and evaluation easier to follow.

Appendix - Page 8:

"Generating lab notes includes a series of binary error detection decisions (i.e., flagging a step as 'correct' or 'wrong'). To ensure the consistency of these decisions, we selected a low sampling temperature of 0.2. While a temperature of 0.0 would offer complete determinism, it is not ideal for this generative task, which involves multiple tokens. In our experience, a temperature of 0.0 occasionally caused the model to enter undesirable generation loops (e.g., repeating the same phrase or word multiple times), resulting in an incomplete or malformed lab note. A low temperature of 0.2 provides the optimal balance: it is sufficiently low to make the error detection consistent and reproducible, while introducing just enough stochasticity to prevent these repetitive generative artifacts."

6. The LLM-as-a-judge approach: was it confirmed that the model's Likert scale corresponds to an accurate semi-quantitative assessment? In other words, is the LLM judge calibrated with respect to the rubrics that were assessed (completeness, technical accuracy, etc)? A sample of LLM judgments could be re-evaluated blindly and the agreement to a human rater calculated by Cohen's Kappa or correlation.

We thank the reviewer for this comment and have now performed the recommended calibration analysis.

Five expert human raters blindly evaluated the ten AI-generated protocols using the same five criteria (completeness, technical accuracy, etc.). This process revealed an important methodological insight: our

initial LLM-as-a-judge method, which calculated average scores from the ratings of every single section and protocol step, did not rate in the same way as humans. This is because a person cannot rate every single step; instead, they naturally provide a more holistic 'overall' score for each criterion. Therefore, to create a valid comparison, we adjusted the 'LLM-as-a-judge' to also provide a single, holistic score per category after its detailed comparison of every step and section. This new approach aligns much better with the human rating process, where a single critical error can (and should) drastically lower the overall score for a category.

We have updated the revised manuscript with the evaluations from both the human raters and the adjusted LLM-as-a-judge, as well as a calibration analysis comparing the human and LLM scores using Spearman's correlation.

Page 2:

“We compared the AI-generated protocols across ten diverse procedures, from simple pipetting to complex mass spectrometer operations to an established lab ground truth. Human raters assigned the AI protocols achieved an average quality score of 4.15 out of 5 based on their completeness, technical accuracy, logical flow, safety instructions, and formatting (see Appendix & Appendix Figure S3).”

Appendix Figure S3: Performance metrics for the protocol generation task.

(A) Composition of the benchmark dataset, categorized into specialized software, specialized equipment, and regular wet-lab protocols. (B) The left panel shows a box plot of the source video durations. The right scatter plot illustrates the relationship between the protocol generation time (in seconds) and the corresponding token cost per generation (in US dollars) for each protocol. (C) Evaluation scores for the generated protocols from five expert human raters (blue) and five LLM-as-a-judge iterations (green) across six categories: completeness, technical accuracy, logical flow, safety instructions, formatting, and an overall quality. The height of each bar represents the average score, and the error bars indicate the standard deviation. (D) Calibration of the "LLM-as-a-judge" against human ratings. Each bar displays the Spearman's rank correlation coefficient (ρ) for a single evaluation criterion. This coefficient quantifies the correlation between the mean scores of five expert human raters and the mean scores from the five LLM-as-a-judge iterations, calculated across the ten generated protocols.

Appendix – Page 8:

"To validate this 'LLM-as-a-judge' approach, we performed a calibration experiment. Five expert human raters were tasked with evaluating the same ten AI-generated protocols against the ground-truth protocols. Mirroring the LLM-judge, the human experts also provided a single, holistic score for each of the five evaluation criteria using the identical 1-5 rating rubric.

We found strong Spearman's correlations (ρ) for all critical scientific criteria, including Technical Accuracy ($\rho \approx 0.91$), Logical Flow ($\rho \approx 0.86$), Safety ($\rho \approx 0.89$), and Overall Quality ($\rho \approx 0.88$) (Appendix Figure S3 D). This analysis also revealed distinct rating profiles: human experts were more stringent on content-related criteria like Completeness and Technical Accuracy, which require expert understanding, whereas the LLM-judge was stricter when comparing the AI-protocol with the ground truth on Formatting (Appendix Figure S3 C). This divergence in focus explains the lack of correlation for the formatting rubric ($\rho \approx 0.05$), confirming the LLM and humans interpreted this specific criterion differently. Overall, this calibration experiment showed that there is a strong correlation between the LLM and human scores, confirming that our LLM judge is well-calibrated."

7. Is the LLM-as-a-judge evaluation sufficiently general to evaluate application of the system in other labs? To what extent? Only for MS proteomics, related disciplines, less related wet lab disciplines?

We aimed to design the LLM-as-a-judge prompt to be as domain-agnostic as possible, as is now specified in the Appendix. We also added a protocol example to the Appendix to make the generation and evaluation process more transparent (Page 9-13).

Appendix - Page 8:

"For automated evaluation, we used an 'LLM-as-a-judge' approach (Zheng et al, 2023), where a separate model compared the AI-generated protocols with manually created ground-truth section by section and step by step. To ensure a general-purpose evaluation applicable to protocols from diverse domains, the LLM-as-a-judge prompt first instructs the model to adopt a domain-independent persona as an "expert evaluator". It then follows a structured reasoning process, directing the model to first read both the ground truth and AI-generated protocols and systematically compare each section (e.g., Title, Abstract, Materials, Procedure steps). Finally,

based on this detailed comparison, the LLM is instructed to provide a single, holistic rating for each protocol on a 1-5 (5: very good) scale across the five evaluation criteria (see Table 1). To calculate the final overall quality score, the scores of the five evaluation criteria were averaged.

8. Could it be an issue that the errors used in the tests were deliberately introduced? How high is the risk that errors which happen in reality will be out of distribution of the artificial errors tested here?

We have now clarified in the manuscript how the curation process for the benchmark mitigates the risk of out-of-distribution errors.

Appendix - Page 13:

“Using the ten protocols, we generated a benchmark dataset of 28 videos in which an expert scientist performed the procedures. Most videos contained intentionally introduced errors, while some were error-free controls. In total, the dataset comprised 421 steps, including 70 deliberate errors. To mitigate the risk of out-of-distribution errors, these 70 errors were curated by expert scientists to mimic common, real-world procedural mistakes observed in our laboratory such as omitted steps or incorrect step execution. While the resulting error frequency of 17% in this dataset is intentionally higher than one would expect from a trained researcher (to facilitate robust model testing), the error types and proportion are representative of those that occur in practice.”

9. The code repository looks solid and passes most of the thresholds for open source projects. However, I notice a lack of testing and continuous integration, which could be problematic if the community involvement the authors call for actually does happen. Without a solid test battery, contributions are very difficult to manage. I recommend adding tests to cover the existing functionality to ensure later additions do not break things. I also recommend thinking about versioning the framework (semantic versioning if possible) for the same reasons. A dedicated documentation that is automatically built from the repository (e.g., using mkdocs material) would be a further step towards sustainability in a community-driven way. Documentation solely in the GitHub README has its limitations.

We have now added a unit test suite achieving 93% coverage and supplemented the README with documentation using Sphinx.

Minor

10. I recommend another round of grammar checks (often regarding commas); for instance, the sentence "We compared ... [,] to an established lab ground truth." is ambiguous due to a missing comma; in other places, there are unnecessary ones.

The manuscript has now been additionally proofread for grammar and comma usage. All noted errors, including the example provided, have been corrected.

2nd Dec 2025

Manuscript number: MSB-2025-13390R

Title: Multimodal AI agents for capturing and sharing proteomics laboratory practice

Dear Matthias,

Thank you again for sending us your revised manuscript. We are now satisfied with the modifications made and I am pleased to inform you that your paper has been accepted for publication.

Kind regards,
Jingyi

Jingyi Hou, PhD
Senior Editor
Molecular Systems Biology
